# Design, Synthesis and Evaluation of New Indolylpyrimidylpiperazines for Gastrointestinal Cancer Therapy

**DOI:** 10.3390/molecules24203661

**Published:** 2019-10-11

**Authors:** Aaron Tan, Maria V. Babak, Gopalakrishnan Venkatesan, Clarissa Lim, Karl-Norbert Klotz, Deron Raymond Herr, Siew Lee Cheong, Stephanie Federico, Giampiero Spalluto, Wei-Yi Ong, Yu Zong Chen, Jason Siau Ee Loo, Giorgia Pastorin

**Affiliations:** 1NUS Graduate School of Integrative Sciences and Engineering, National University of Singapore, Centre for Life Sciences, #05-01, 28 Medical Drive, Singapore 117456, Singapore; aarontan@u.nus.edu; 2Department of Pharmacy, National University of Singapore, Singapore 119260, Singapore; maria.babak@nus.edu.sg (M.V.B.); phagove@nus.edu.sg (G.V.); scifryc@nus.edu.sg (C.L.); csccyz@nus.edu.sg (Y.Z.C.); 3Institut für Pharmakologie und Toxikologie, Universität Würzburg, 97078 Würzburg, Germany; klotz@toxi.uni-wuerzburg.de; 4Department of Pharmacology, National University of Singapore, Singapore 117600, Singapore; phcdrh@nus.edu.sg; 5Department of Pharmaceutical Chemistry, School of Pharmacy, International Medical University, 126 Jalan Jalil Perkasa 19, Bukit Jalil, Kuala Lumpur 57000, Malaysia; 6Dipartimento di Scienze Chimiche e Farmaceutiche, Università degli Studi di Trieste, 34127 Trieste, Italy; sfederico@units.it (S.F.); spalluto@units.it (G.S.); 7Department of Anatomy, National University of Singapore, Singapore 119260, Singapore; antongwy@nus.edu.sg; 8School of Pharmacy, Faculty of Health and Medical Sciences, Taylor’s University, 1, Jalan Taylors, Subang Jaya, Selangor 47500, Malaysia

**Keywords:** gastrointestinal cancer, hA_3_AR, partial agonists, indolylpyrimidylpiperazines

## Abstract

Human A_3_ adenosine receptor hA_3_AR has been implicated in gastrointestinal cancer, where its cellular expression has been found increased, thus suggesting its potential as a molecular target for novel anticancer compounds. Observation made in our previous work indicated the importance of the carbonyl group of amide in the indolylpyrimidylpiperazine (IPP) for its human A_2A_ adenosine receptor (hA_2A_AR) subtype binding selectivity over the other AR subtypes. Taking this observation into account, we structurally modified an indolylpyrimidylpiperazine (IPP) scaffold, **1** (a non-selective adenosine receptors’ ligand) into a modified IPP (mIPP) scaffold by switching the position of the carbonyl group, resulting in the formation of both ketone and tertiary amine groups in the new scaffold. Results showed that such modification diminished the A_2A_ activity and instead conferred hA_3_AR agonistic activity. Among the new mIPP derivatives (**3**–**6**), compound **4** showed potential as a hA_3_AR partial agonist, with an E_max_ of 30% and EC_50_ of 2.89 ± 0.55 μM. In the cytotoxicity assays, compound **4** also exhibited higher cytotoxicity against both colorectal and liver cancer cells as compared to normal cells. Overall, this new series of compounds provide a promising starting point for further development of potent and selective hA_3_AR partial agonists for the treatment of gastrointestinal cancers.

## 1. Introduction

Gastrointestinal (GI)-related cancers are pathological conditions that account for at least 30% of cancer-related deaths worldwide [1]. Among them, liver cancer has shown an increasing incidence, which is projected to rise by 35% by 2030 [2]. Colorectal cancer is also becoming a predominant cancer worldwide, with 746,300 and 614,300 new cases reported in 2012 among men and women, respectively [3]; its incidence and death rates are projected to rise by 60% by 2030 [4]. This projection is indicative of colorectal cancer becoming the cancer with the highest incidence and the second most prevalent cause of cancer-related deaths by 2030.

The current chemotherapeutic options for colorectal cancers are limited. Nonetheless, a study conducted by Bar-Yehuda et al. showed that a human A_3_ adenosine receptor (hA_3_AR)-selective agonist, CF101 (IB-MECA), was able to enhance the chemotherapeutic efficacy of 5-fluorouracil in a colon carcinoma murine model. Sakowicz-Burkiewicz et al. also observed that the treatment of HCT-116 human colorectal carcinoma cells with CF101 had resulted in an increase in apoptosis and necrosis of the cancer cells [5]. Aphase II clinical trial on human colon carcinoma patients showed CF101 exhibiting disease stabilization potential in 35% of the patients [6]. In addition, CF101 also exhibited a myeloprotective effect in the murine model [7], suggesting improved tolerability and decreased side effects. From these studies, it emerges that the activation of the hA_3_AR clearly plays a crucial role in tumour progression: this receptor is overexpressed in several pathological conditions [8] such as colorectal cancer [9,10], breast cancer [10,11], brain cancer [12] and leukaemia [13]. Patients suffering from colorectal cancer were also reported to have a higher expression level of the hA_3_AR at the tumour site as compared to adjacent, remote, and healthy colon mucosa. Furthermore, the level of upregulation of hA_3_AR was found to be directly correlated to the disease severity [9], making it a representative biomarker for colorectal cancer.

Similarly, poor prognosis is associated with advanced stage liver cancer patients, where drugs such as doxorubicin and cisplatin do not exhibit improved survival rate (as observed in various phase III clinical trials) and are accompanied with significant toxicity [14]. Hence, many research groups are exploring new molecular targets for advanced stages of liver cancer. One current phase II clinical trial is investigating the use of CF102 (2-Cl-IB-MECA), a hA_3_AR-selective agonist, to improve the overall survival rate of advanced liver cancer patients while maintaining a low toxicity profile. Earlier phase I/II clinical trials on 18 advanced stage liver cancer patients showed that a patient who developed metastases prior to CF102 treatment had achieved a complete metastasis regression during the three-month treatment with CF102 [15]. The success of both CF101 and CF102 demonstrate the potential for selective hA_3_AR agonists.

In our previous work, we had developed a hA_2A_AR-selective ligand, **1**. An observation based on such work indicated that a simple chemical modification in the scaffold of **1** was able to convert a selective hA_2A_AR ligand with K_i_ hA_2A_AR = 8.71 μM and selectivity hA_2A_AR/hA_3_AR > 12 into a non-selective ligand, **2**, with K_i_ hA_2A_AR = 34.40 μM and K_i_ hA_3_AR = 39.80 μM (Figure 1). In other words, the simple removal of the carbonyl group of amide in **1** had led to the loss of binding selectivity towards the hA_2A_AR [16].

In continuation to the previous studies, we aimed to investigate the effect of switching the position of the carbonyl group on indolylpyrimidylpiperazine (IPP) towards the activity at both A_2A_ and A_3_ receptors in our present work. Hence, we carried out structural modifications of IPP (**1**) by shifting the carbonyl group to the position adjacent to the indole ring as shown in the modified IPP (mIPP), **3**, producing both a ketone group and a tertiary amine group in the new scaffold. It is postulated that such modification would eventually diminish the A_2A_ activity while gaining activity at the A_3_ receptor. Based on the mIPP scaffold, **3**, we introduced different substituents on the 7th position of the indole ring, generating the series of compounds, **4**–**6** (shown in Figure 2).

## 2. Results and Discussion

### 2.1. Chemistry

Compound **3** was synthesized based upon the synthetic scheme (Scheme 1), in which indole was treated dropwise with chloroacetylchloride in the presence of pyridine for an hour at 60 °C in dioxane to give 2-chloro-1-(1H-indol-3-yl)ethanone (**7**), with a yield of 50% [17].

The same synthetic scheme could not be applied for the synthesis of compounds **4**–**6**, as the substituted chloroacetylindoles could not be produced using the same reaction condition as for the synthesis of compound **7**.

Compounds **4** and **5** were synthesized based on the synthetic scheme (Scheme 2), where both 7-methoxyindole and 7-chloroindole were treated with chloroacetylchloride in the presence of 1,8-diazabicyclo(5.4.0)undec-7-ene (DBU) under a pressurized reaction condition in dichloroethane (DCE) at 90 °C for a reaction period of 2–5 days to afford 2-chloro-1-(7-methoxy-1H-indol-3-yl)ethanone (**9**) and 2-chloro-1-(7-chloro-1H-indol-3-yl)ethanone (**10**) in 23% and 11% yield, respectively [18].

Compound **6** was obtained through the Vilsmeier‒Haack acylation for synthesis of compound **12**, 2-chloro-1-(7-nitro-1H-indol-3-yl)ethenone, as presented in Scheme 3. 2-Chloro-*N,N*-dimethylacetamide (**11**) was first synthesized from chloroacetyl chloride and dimethylamine hydrochloride salt in the presence of triethylamine (TEA) and was subsequently used in a Vilsmeier‒Haack acylation of 7-nitroindole in the presence of phosphoryl oxychloride at room temperature for 18 h to give 2-chloro-1-(7-nitro-1H-indol-3-yl)ethenone (**12**) with a yield of 17% [19].

To minimize the formation of 2-[4-(4,6-dimethylpyrimidin-2-yl)piperazin-1-yl]-4,6-dimethylpyrimidine, five equivalents of piperazine were reacted with one equivalent of 2-chloro-4,6-dimethylpyrimdine in a solvent mixture of tetrahydrofuran (THF) and water with a ratio of 1:4 at room temperature for 24 h in the presence of base to give 4,6-dimethyl-2-(piperazin-1-yl)pyrimidine (**8**) with a yield of 96% (Scheme 1) [20].

Compounds **3**–**6** were then obtained from reaction between compound **8** and their respective chloroacetylindoles **7**, **9**, **10** and **12** in the presence of sodium iodide and base under reflux for 24 h, with respective yields of 38%, 77%, 59% and 69% [21]. 

### 2.2. Biological Evaluation

#### 2.2.1. Functional Activity Studies at hA_2A_AR and hA_3_AR

Functional activity of compounds **3**–**6** was determined using the transforming growth factor alpha (TGF_α_) shedding assay. This recently-developed assay utilizes an alkaline phosphatase-TGF_α_ fusion protein reporter system to quantitatively evaluate specific GPCR-mediated G protein activation (shown in Figure 3) of test compounds [22].

The TGF_α_ shedding assay results of compounds **3**–**6** on hA_3_AR (shown in Figure 4), summarized in Table 1, revealed that the compounds exhibited agonistic activity at the A_3_ receptor. Compound **4** is the most potent hA_3_AR agonist among the four compounds **3**–**6**, with an EC_50_ of 2.89 ± 0.55 μM. It is also a partial hA_3_AR agonist, with an E_max_ of 31%, as compared to the efficacy of adenosine. Similarly, compounds **5** and **6** are also weak hA_3_AR agonists, with respective EC_50_ values of 13.4 ± 2.96 μM and 33.2 ± 16.50 μM but have higher efficacy of 77% and 72%, respectively. On the other hand, compound **3** is essentially inactive against the hA_3_AR. 

The TGF_α_ shedding assay at hA_2A_AR was also performed on the compounds **3**–**6**. Results (Figure 5) showed that compounds **3**–**6** are all essentially inactive against the human A_2A_AR, confirming that compounds **4**–**6** are all human A_3_AR-selective partial agonists with respect to the hA_2A_AR.

TGF_α_ shedding assay results of compounds **3**–**6** on vector-transfected HEK293 cells, with neither human A_2A_AR nor human A_3_AR expressed (as shown in Figure 6) confirmed that the activity observed in the TGF_α_ shedding assay on hA_3_AR-expressing HEK293 cells is due to the action of the compounds on the hA_3_AR, eliminating eventual external interferences in the HEK293 cells.

The shifting of carbonyl group position in the mIPP scaffold has led to significant reduction of the hA_2A_AR activity, while simultaneously increasing the hA_3_AR activity and selectivity as A_3_-selective partial agonists. Of note, previous studies had shown that hA_3_AR-selective full agonists could cause rapid desensitization via β-arrestin2 trafficking of the hA_3_AR upon receptor activation [23,24]. Consistently, Bar-Yehuda et al. reported that a reduction in the hA_3_AR expression level was observed in A_3_ full agonist-treated hepatocellular carcinoma (HCC) tumour-bearing rats as compared to vehicle-treated HCC tumour-bearing rats, indicative of desensitization and reduced expression of hA_3_AR [25]. In a study by Gao et al., the partial hA_3_AR-selective agonist, MRS541, was demonstrated to have a lower β-arrestin translocation response of 30.8 ± 1.6% in comparison to the full A_3_AR-selective agonists, IB-MECA and 2-Cl-IB-MECA, which have a high β-arrestin translocation response of 100 ± 7% and 102 ± 13%, respectively [26]. It can be deduced from these findings that hA_3_AR-selective partial agonists would have a lower tendency of triggering rapid receptor trafficking and desensitization of hA_3_AR, particularly in cancers that overexpress the A_3_ receptors.

Among the four compounds, compound **4** was subsequently chosen for further evaluation of its anti-proliferative activity in human colorectal and liver cancer cell lines. This is because it is the most potent among the new mIPP derivatives, with an EC_50_ of 2.89 ± 0.55 μM at the hA_3_AR.

#### 2.2.2. Cytotoxicity Assays of Compound **4** against Human Colorectal Cancer Cell Lines, a Normal Colon Cell Line and a Liver Cancer Cell Line.

Human colorectal carcinoma cell lines, HCT-116 and Caco-2, were used as models for colorectal cancer cells overexpressing hA_3_AR. Human normal colon cell line CCD-18Co was used as a model for normal colon cells that express normal levels of hA_3_AR. The HCT-116, Caco-2 and CCD-18Co cell lines were all cultured in accordance with the ATCC protocols.

Cytotoxicity assay results, as shown in Figure 7, indicated that compound **4** is approximately two times more cytotoxic towards the colorectal cancer cell lines, i.e., the HCT-116 cell line (IC_50_ = 84 ± 9 μM) and the Caco-2 cell line (IC_50_ = 77 ± 10 μM), than the normal colon cell line, the CCD-18Co cell line (IC_50_ = 176 ± 48 μM). Hence, compound **4** is cytotoxic-selective for the colorectal cancer cells over its normal counterpart.

As both healthy and cancerous livers have high expression of hA_3_AR, which is remarkably higher than that of colorectal cancer cells, human hepatocellular carcinoma cells (HepG2) were used to test if compound **4** is more potent against hepatocellular carcinoma cells versus human colorectal carcinoma cells. The HepG2 cell line was also cultured in accordance with the ATCC protocol. 

Based on the results obtained, compound **4** showed higher cytotoxicity against the HepG2 cell line (IC_50_ = 30 ± 7 μM) (Table 2), a hepatocellular carcinoma cell line, than against the colorectal cancer cell lines. It also exhibited cytotoxicity of the same order of magnitude to 2-Cl-IB-MECA (Figure 8), a highly potent hA_3_AR-selective agonist. This result suggests that compound **4**, a hA_3_AR-selective partial agonist, can exert the same cytotoxicity as the hA_3_AR-selective full agonist, 2-Cl-IB-MECA. This is probably due to the rapid hA_3_AR desensitization upon exposure of the cancer cell lines to the hA_3_AR full agonist, 2-Cl-IB-MECA, as a previous study by Fishman et al. showed significant downregulation of hA_3_AR protein level after only 18 h of IB-MECA treatment on PC-3 prostate carcinoma cells [27]. Compound **4** might have a lower tendency of causing hA_3_AR desensitization, which would account for the similar order of magnitude of cytotoxicity as 2-Cl-IB-MECA against the cancer cell lines. However, this would need to be further confirmed with a biological study on compound **4** to assess its effect towards A_3_AR trafficking by β-arrestin.

### 2.3. UV-Vis Stability Study of Compound ***4***

UV-vis stability study of compound **4** showed that it exhibited excellent stability in water, Roswell Park Memorial Institute (RPMI)-1640 culture medium and Dulbecco’s Modified Eagle’s medium (DMEM) culture medium over 72 h, as indicated by the absence of isosbestic points within the UV-vis spectrum (Appendix A).

### 2.4. Induced-Fit Docking of Compound ***1*** and Compounds ***3***–***6*** in hA_3_AR Homology Model

Induced-fit docking of compound **1** and compounds **3**–**6** was conducted to rationalize the hA_3_AR agonistic activity of compounds **3**–**6** based on the hA_3_AR homology model built from the active state hA_2A_AR crystal structure. 

As observed in Figure 9A, the induced-fit docking pose for compound **1** indicated a distinct pose compared to that of compounds **4**–**6**. We observed the absence of a crucial hydrogen bonding interaction between the carbonyl group of the amide and Asn250^6.55^ of the hA_3_AR homology model in compound **1**; conversely, the induced-fit docking pose for compound **4** showed the hydrogen bonding interaction between the carbonyl group adjacent to the indole ring and the Asn250^6.55^. This docking result supports our postulation that the shift in position of the carbonyl group from the amide to the position adjacent to the indole ring would confer binding affinity for the hA_3_AR, which has been then proven by the gain of hA_3_ activity in compound **4**.

Generally, the induced-fit docking poses for compounds **3**–**6** well overlapped with each other in the active site of the hA_3_AR homology model (shown in Figure 9B), adopting similar conformations with the formation of crucial hydrogen bonding interaction between the carbonyl group of the compounds and Asn250^6.55^ as well as the π-π stacking interaction between the indolyl ring of the compounds and Phe168.

In particular, an interesting observation was made from the induced-fit docking results of compounds **3**–**6**. The Trp243^6.48^ was found to adopt a different degree of rotation for compounds **4**–**6** as compared to that of compound **3** (shown in Figure 9C). Previous molecular modelling studies with a hA_3_AR homology model built from the hA_2A_AR crystal structure also reported similar observation of rotation of the Trp243^6.48^ residue from its inactive “vertical” conformation to the active “horizontal” conformation upon agonist binding [28,29,30,31]. In fact, a site-directed mutagenesis study reported in Gao et al. demonstrated that Trp243^6.48^ is a residue crucial for the A_3_ receptor activation [32,33,34]. This residue might act as a switch in TM6-mediated structural transition from the resting to the active state of the A_3_ receptor upon binding of agonist. The difference in rotation of Trp243 residue observed between compound **3** and compounds **4**–**6** is in concordance with our findings from the TGF_α_ shedding assay, whereby compound **3** was found to be inactive against hA_3_AR, while compounds **4**–**6** showed relatively higher hA_3_AR agonistic activity. 

## 3. Materials and Methods 

### 3.1. Chemistry

General ^1^H and ^13^C nuclear magnetic resonance (NMR) spectra (DMSO-d_6_) were recorded on a Bruker DPX Ultrashield NMR (400 MHz). Chemical shifts are reported as parts per million (δ) relative to the solvent peak. Coupling constants (*J*) are reported in hertz (Hz). High Performance Liquid Chromatography (HPLC) analysis was conducted with Agilent HP-1100 system equipped with a diode array detector (DAD). Reactions were checked with thin layer chromatography (TLC) (Merck precoated 60F_254_ plates). Flash chromatography was performed on silica gel 60 (70-230 mesh, Merck). Unless otherwise stated, materials were obtained from commercial suppliers and used without further purification. All anhydrous and technical-grade solvents were also obtained from commercial suppliers and used without further purification or drying.

**2-Chloro-1-(1H-indol-3-yl)ethanone (7)**: Pyridine (1 equiv.) was added to a solution of indole (1 equiv.) in 20 mL of dry dioxane under argon and the mixture was warmed to 60 °C. Chloroacetyl chloride (1 equiv.) was added dropwise over 10 min and the reaction mixture was allowed to stir at 60 °C for 1 h. The reaction mixture was poured into 40 mL of iced water and extracted with 40 mL of ethyl acetate (EA) twice. The organic layer was washed with brine before drying with anhydrous sodium sulphate. The mixture was concentrated in vacuo and crudes were dry-loaded onto silica gel and purified with column chromatography (EA: Hexane = 1:100 to 1:4) to give the desired product, 3-chloroacetylindole as a brownish solid in 50% yield. ^1^H NMR (400 MHz, DMSO-d_6_): δ = 12.12 (s, 1 H), 8.43 (d, *J* = 3.24 Hz, 1 H), 8.17–8.15 (m, 1 H), 7.51–7.49 (m, 1 H), 7.27–7.23 (m, 1 H), 7.23–7.20 (m, 1 H), 4.87 (s, 2 H). ^13^C NMR (DMSO-d_6_): δ = 186.16, 136.61, 134.75, 125.39, 123.19, 122.15, 121.15, 113.61, 112.33, 46.38.

**4,6-Dimethyl-2-(piperazin-1-yl)pyrimidine (8)**: To a stirred solution of piperazine (5 equiv.) and potassium carbonate (5 equiv.) in 24 mL of deionized water, 2-chloro-4,6-dimethylpyrimidine (1 equiv.) in 6 mL of tetrahydrofuran was added. The reaction mixture was stirred at room temperature for 24 h. After 24 h, tetrahydrofuran was evaporated in vacuo and the aqueous layer was extracted with 25 mL of EA three times. The combined organic layer was dried with anhydrous sodium sulphate and dry-loaded onto silica gel. Crudes dry-loaded on silica gel were purified by flash chromatography (EA: Hexane = 1:100–100:1) to give 4,6-dimethyl-2-(piperazin-1-yl)pyrimidine as a white solid in 96% yield. ^1^H NMR (400 MHz, DMSO-d_6_): δ = 6.35 (s, 1 H), 3.63 (t, *J* = 5.00 Hz, 4 H), 2.69 (t, *J* = 4.88 Hz, 4 H), 2.20 (s, 6H). ^13^C NMR (DMSO-d_6_): δ = 166.47, 161.28, 108.37, 45.55, 44.46, 23.69.

**General procedure for direct acylation to obtain 7-methoxy and 7-chloro substituted 3-chloroacetylindole (9–10)**. 7-substituted indoles (1 equiv.) was weighed in a microwave vial and the microwave vial was capped after adding the stir bar before purging with argon. 5 mL of dichloroethane (DCE) was added to the microwave vial to dissolve the 7-substituted indoles. DBU (1 equiv.) was added to the reaction mixture and heated in an oil bath to 90 °C. Upon reaching 90 °C, chloroacetyl chloride (1.1 equiv.) was added in a single portion into the microwave vial via syringe. The reaction was stirred and monitored with TLC hourly until product spot was observed and qualitative yield was obtained, before heating was stopped. The reaction mixture upon cooling was added to a water-methanol mixture of 6:1 under vigorous stirring to quench the reaction. The quenched reaction was allowed to stir for an additional 1 h and solid crude product was filtered with a sintered glass filter funnel before washing with water. The crude product was allowed to dry in an oven before dissolving in methanol. The dissolved crude product was dry loaded onto silica gel and was purified with column chromatography (EA: Hexane = 1:100–1:4) to give the pure 7-substituted 3-chloroacetylindole.

**2-Chloro-1-(7-methoxy-1H-indol-3-yl)ethanone (9):** Brownish solid, yield: 23%. ^1^H NMR (400 MHz, DMSO-d_6_): δ = 12.27 (s, 1 H), 8.32 (d, *J* = 3.28 Hz, 1 H), 7.74–7.72 (d, *J* = 7.92 Hz, 1 H), 7.14 (t, *J* = 7.92 Hz, 1 H), 6.82 (d, *J* = 7.44 Hz, 1 H), 4.88 (s, 2 H), 3.94 (s, 3H). ^13^C NMR (DMSO-d_6_): δ = 186.20, 146.27, 133.96, 126.94, 126.59, 122.95, 114.10, 113.64, 103.86, 55.32, 46.48.

**2-Chloro-1-(7-chloro-1H-indol-3-yl)ethanone (10):** Yellowish solid, yield: 11%. ^1^H NMR (400 MHz, DMSO-d_6_): δ = 12.50 (s, 1 H), 8.52–8.51 (d, *J* = 3.24 Hz, 1 H), 8.13–8.11 (d, *J* = 7.88 Hz, 1 H), 7.36–7.33 (dd, *J* = 7.64 Hz, 0.68 Hz, 1 H), 7.23 (t, *J* = 7.80 Hz, 1 H), 4.94 (s, 2 H). ^13^C NMR (DMSO-d_6_): δ = 186.42, 135.55, 133.52, 127.27, 123.29, 122.79, 120.09, 116.62, 114.43, 46.54.

**Procedure for Vilsmeier‒Haack acylation to obtain 7-nitro substituted 3-chloroacetylindole (12).** Chloroacetyl chloride (3 equiv.) was added dropwise to a stirring mixture of dimethylamine hydrochloride (1 equiv.) and triethylamine (1.1 equiv.) in an ice bath. The reaction was allowed to stir further in the ice bath for 3 h before 40 mL of water was added for quenching the reaction. The aqueous layer was extracted with 50 mL of dichloromethane (DCM) twice and the combined organic layer was washed three times with 50mL of 1M hydrochloric acid. The organic layer was concentrated in vacuo before being dispersed in diethyl ether and stirred overnight. Solid precipitate was filtered away and the organic layer was reconcentrated in vacuo to afford the purified 2-chloro-*N,N*-dimethylacetamide (**11**) as a liquid.

7-nitroindole (1 equiv.) was dissolved in a neat solution of **11** (5 equiv.). The mixture was stirred under argon in an ice bath. Phosphorus oxychloride (2 equiv.) was added dropwise into the reaction mixture in the ice bath. The reaction mixture was allowed to stir overnight at room temperature. The reaction was neutralized with 4 mL of 2M aqueous sodium hydroxide solution. The aqueous layer was extracted with 10mL of EA three times. Combined organic layers were washed with 30 mL saturated sodium bicarbonate twice. The organic layer was dried over anhydrous sodium sulphate and concentrated in vacuo. Residue obtained was purified with column chromatography (EA: Hexane = 1:100–1:4) to give pure 2-chloro-1-(7-nitro-1H-indol-3-yl)ethanone (**12**).

**2-Chloro-*N,N*-dimethylacetamide (11):** Dark brownish liquid, yield: 62%. ^1^H NMR (400 MHz, CDCl_3_): δ = 3.86 (s, 1 H), 2.81 (s, 3 H), 2.67 (s, 3 H). ^13^C NMR (CDCl_3_): δ = 166.06, 40.84, 36.92, 35.31.

**2-Chloro-1-(7-nitro-1H-indol-3-yl)ethanone (12):** Brownish solid, yield: 17%. ^1^H NMR (400 MHz, DMSO-d_6_): δ = 12.73 (s, 1 H), 8.64–8.62 (dd, *J* = 7.84 Hz, 0.92 Hz, 1 H), 8.57 (d, *J* = 2.08 Hz, 1 H), 8.24–8.21 (dd, *J* = 8.04 Hz, 0.90 Hz, 1 H), 7.47 (t, *J* = 7.96 Hz, 1 H), 5.02 (s, 2 H). ^13^C NMR (DMSO-d_6_): δ = 186.91, 137.11, 133.39, 129.35, 129.08, 128.82, 122.20, 120.18, 114.26, 46.80.

**General procedure for synthesis of mIPP and its derivatives (3–6).** Chloroacetylindole (1 equiv.) was added to a stirred mixture of 4,6-dimethyl-2-(piperazin-1-yl)pyrimidine (1 equiv.), potassium carbonate (3 equiv.) and sodium iodide (1 equiv.) in 60 mL of acetonitrile. The reaction mixture was stirred under reflux for 4 h. Upon completion of the reaction determined by TLC analysis, the reaction mixture was cooled to room temperature and acetonitrile was removed in vacuo to give a dry residue. The dry residue was washed with water to remove the salts and the precipitate was dissolved in methanol and dry-loaded onto silica gel. Crude dry-loaded on silica gel were purified by column chromatography (EA: Hexane = 1:100–3:1) to give mIPP and its derivatives.

**2-(4-(4,6-Dimethylpyrimidin-2-yl)piperazin-1-yl)-1-(1H-indol-3-yl)ethanone (3)**: White solid, yield: 38%, mp = 165 °C (decomp): ^1^H NMR (400 MHz, DMSO-d_6_) δ11.91 (s, 1H), 8.49 (s, 1H), 8.20–8.18 (m, 1H), 7.48–7.46 (m, 1H), 7.23–7.20 (m, 1H), 7.19–7.16 (m, 1H), 6.40 (s, 1H), 3.76–3.74 (m, 4H), 3.66 (s, 2H), 2.59–2.57 (m, 4H), 2.22 (s, 6H).^13^C NMR (DMSO-d_6_): δ = 192.37, 166.63, 161.17, 136.28, 134.19, 125.60, 122.74, 121.72, 121.32, 115.23, 112.07, 108.72, 64.99, 52.79, 43.31, 23.69. HPLC analysis of compound purity by two solvent systems: ACN:H_2_O = 70:30 (98%), MeOH:H_2_O = 70:30 (99%). 

**2-(4-(4,6-Dimethylpyrimidin-2-yl)piperazin-1-yl)-1-(7-methoxy-1H-indol-3-yl)ethanone (4)**: White solid, yield: 77%, mp = 162 °C (decomp): ^1^H NMR (400 MHz, DMSO-d_6_) δ12.05 (s, 1H), 8.36 (d, *J* = 3.16 Hz, 1H), 7.76 (d, *J* = 7.96 Hz, 1H), 7.12–7.08 (m, 1H), 6.80-6.78 (m, 1H), 6.39 (s, 1H), 3.93 (s, 3H), 3.74–3.72 (m, 4H), 3.64 (s, 2H), 2.58-2.55 (m, 4H), 2.21 (s, 6H).^13^C NMR (DMSO-d_6_): δ = 192.43, 166.64, 161.17, 146.20, 133.40, 127.16, 126.28, 122.53, 115.74, 113.91, 108.73, 103.49, 65.01, 55.29, 52.75, 43.30, 23.68. HPLC analysis of compound purity by two solvent systems: ACN:H_2_O = 70:30 (95%), MeOH:H_2_O = 70:30 (95%).

**2-(4-(4,6-Dimethylpyrimidin-2-yl)piperazin-1-yl)-1-(7-chloro-1H-indol-3-yl)ethanone (5)**: White solid, yield: 59%, mp = 163 °C (decomp): ^1^H NMR (400 MHz, DMSO-d_6_) δ12.31 (s, 1H), 8.56 (d, *J* = 2.84 Hz, 1H), 8.17–8.15 (m, 1H), 7.32–7.30 (dd, *J* = 7.60 Hz, 0.86 Hz, 1H), 7.20 (t, *J* = 7.84 Hz, 1H), 6.39 (s, 1H), 3.75–3.72 (m, 4H), 3.69 (s, 2H), 2.59–2.56 (m, 4H), 2.22 (s, 6H).^13^C NMR (DMSO-d_6_): δ = 192.62, 166.61, 161.13, 134.99, 133.17, 127.51, 122.88, 122.36, 120.27, 116.43, 115.97, 108.72, 64.97, 52.72, 43.28, 23.68. HPLC analysis of compound purity by two solvent systems: ACN:H_2_O = 70:30 (89%), MeOH:H_2_O = 70:30 (89%). Purification by prep-HPLC was conducted to obtain just the desired product (98% purity).

**2-(4-(4,6-Dimethylpyrimidin-2-yl)piperazin-1-yl)-1-(7-nitro-1H-indol-3-yl)ethanone (6)**: White solid, yield: 69%, mp = 165 °C (decomp): ^1^H NMR (400 MHz, DMSO-d_6_) δ12.56 (s, 1H), 8.68 (d, *J* = 7.36 Hz, 0.92 Hz, 1H), 8.65 (d, *J* = 3.00 Hz, 1H), 8.21 (dd, *J* = 8.04 Hz, 0.90 Hz 1H), 7.47–7.43 (t, *J* = 7.96 Hz, 1H), 6.40 (s, 1H), 3.78 (s, 2H), 3.76 (dd, *J* = 4.84 Hz, 0.42 Hz, 4H), 2.61–2.59 (m, 4H), 2.22 (s, 6H).^13^C NMR (DMSO-d_6_): δ = 166.64, 161.12, 136.62, 133.21, 129.53, 129.33, 128.45, 121.78, 119.80, 115.61, 108.78, 65.06, 52.68, 43.20, 23.68. HPLC analysis of compound purity by two solvent systems: ACN:H_2_O = 70:30 (95%), MeOH:H_2_O = 70:30 (95%).

### 3.2. Biological Evaluation

#### 3.2.1. TGF_α_ Shedding Assays for hA_2A_AR and hA_3_AR

Expression vectors (250 ng of AP-TGF_α_ plasmid, 100 ng of hA_2A_AR/hA_3_AR plasmid and 50 ng of G_α_ protein) were transfected into HEK293 cells using Lipofectamine 2000 (1.25 μL/well) in 12-well plates. Transfection of siRNA (final concentration: 10 nM) was conducted with Lipofectamine RNAiMAX (1 μL/well), 48 h before assay to prevent interference from other signaling pathways. Transfected cells were detached with brief treatment with phosphate-buffered saline (PBS) containing 0.05% trypsin and 0.52 mM ethylenediaminetetraacetic acid (EDTA). The cell suspension was pelleted via centrifugation (190 *g*, 5 min). Trypsin was removed from the cells via resuspension in PBS, followed by 5 min of incubation at room temperature. Cells were subsequently pelleted with centrifugation (190 *g*, 5 min) and resuspended in Hank’s Balanced Salt Solution (HBSS) containing 5 mM 4-(2-hydroxyethyl)-1-piperazineethanesulfonic acid (HEPES) (pH 7.4). Resuspended cells were plated in 90 μL/well in 96-well plate and incubated at 37 °C in 5% CO_2_ for 30 min. 10 μL of 10X concentration of compounds were added and incubated for 1 h at 37 °C in 5% CO_2_. Plates were centrifuged at 190 *g* for 2 min. 160 µL per well of conditioned medium (10 mM p-NPP, 40 mM Tris-HCl (pH 9.5), 40 mM NaCl & 10 mM MgCl_2_) was transferred into a conditioned medium plate and 80 µL per well of conditioned medium was added to the test compound plate. Absorbance at 405 nm (OD405) of both plates was read before and after a 1-h incubation at 37 °C using a microplate reader [22].

#### 3.2.2. Cytotoxicity Assays

The cytotoxicity of compound **4** was determined by 3-(4,5-dimethylthiazol-2-yl)-2,5-diphenyltetrazolium bromide (MTT) assay. The cells were harvested from culture flasks by trypsinisation and seeded into 96-well microculture plates (Costar, Corning, USA) at the seeding density of 6000 cells per well or 10,000 per well (CCD18Co). After the cells were allowed to resume exponential growth for 24 h, they were exposed to compound **4** or 2-Cl-IB-MECA in media for 72 h. The drugs were diluted in complete medium at the desired concentration, and 100 μL of the drug solution was added to each well and serially diluted to other wells. After exposure for 72 h, drug solutions were replaced with 100 μL of MTT in complete media (5 mg ml^−1^) and incubated for an additional 1 h. Subsequently, the medium was aspirated and the purple formazan crystals formed in viable cells were dissolved in 100 μL of dimethyl sulfoxide (DMSO) per well. Optical densities were measured at 570 nm with a microplate reader. The quantity of viable cells was expressed in terms of treated/control (T/C) values by comparison to untreated control cells, and 50% inhibitory concentrations (IC_50_) were calculated from concentration-effect curves by interpolation using GraphPad Prism (Version 5.0, GraphPad Software, USA). Evaluation was based on means from at least three independent experiments, each comprising six replicates per concentration level.

### 3.3. UV-Vis Stability Assay

Stock solution of 50 μM of compound **4** was prepared in DMSO. Two quartz cuvettes were soaked in 70% ethanol in water for 1h. The quartz cuvettes were subsequently left to dry in a biological safety cabinet. When both quartz cuvettes were dry, the quartz cuvettes, i.e., sample cuvette and blank cuvette, were filled with water or culture media (RPMI-1640 or DMEM) and parafilmed before being transferred into a UV-vis spectrophotometer (USA). The UV-vis spectrophotometer was auto-zeroed and the background was scanned from wavelengths of 200 nm to 800 nm. The sample cuvette was removed from the UV-vis spectrophotometer and transferred into the biological safety cabinet. Compound **4** was dissolved at a final concentration of 50 μM in 1.5 mL of water, RPMI or DMEM containing 10% FBS (without phenol red). The UV−vis profiles of the samples were monitored using UV-vis over 72 h at 1 h intervals from 200 to 800 nm.

### 3.4. Molecular Modeling

#### 3.4.1. Homology Modelling and Protein Preparation

The amino acid sequence of hA_3_AR was obtained from the UniProt Knowledgebase (P0DMS8) and was aligned to hA_2A_AR using the structural alignment specified in GPCRdb [35]. The N-terminal and C-terminal were truncated (residues 1‒8 and 305‒318). The hA_3_AR homology model was then constructed using Prime [36], based on the crystal structure of the active-state hA_2A_AR bound to the selective agonist UK-432,097 as a template (PDB ID: 3QAK) [37]. The disulphide bridge between Cys83^3.25^ and Cys166^45.50^ in hA_3_AR was conserved. The stereochemical quality of the hA_3_AR homology model was assessed using PROCHECK [38] (Appendix A). The homology model was subsequently prepared for docking by assigning appropriate protonation states, which were generated at pH 7.4, and by optimizing the hydrogen bond networks. The resulting structure was subjected to energy minimization by using the OPLS 2005 force field. 

#### 3.4.2. Ligand Docking Studies

In-depth computational studies were conducted with Schrödinger Maestro using induced-fit docking protocol. Ligands were prepared for docking via LigPrep [39]. Ligand protonation states were generated at pH 7.4 using Epik [40]. The docking was then performed using the Induced Fit Docking protocol, with the docking grid centred on the hA_3_AR active site [41].

## 4. Conclusions

The present work confirmed our postulation that the structural modification of the IPP scaffold through shifting of the carbonyl group to the position adjacent to the indole ring would diminish the A_2A_ activity while gaining activity at the A_3_ receptor. Among these newly synthesized mIPP derivatives, compound **4** was shown as a hA_3_AR-selective partial agonist with an EC_50_ of 2.89 ± 0.55 µM and E_max_ of 30% that exhibited some extent of selectivity for cancer cells over normal cells. Furthermore, the molecular modelling studies suggest that the difference in rotation of Trp243 residue might account for dissimilarity in agonistic activity observed among the new mIPP derivatives, indicating its role in A_3_ receptor activation. Further structural optimizations of compound **4** would be deemed useful in developing more potent hA_3_AR-selective partial agonists in the future.

In addition, relevant biological studies can be conducted to determine if the hA_3_AR partial agonist possesses a reduced tendency of causing desensitization of the hA_3_AR in cancer cells and reduced non-selective actions towards normal liver tissues that are intrinsically expressing high levels of hA_3_AR.

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
