# Peer review of "Design, Synthesis and Evaluation of New Indolylpyrimidylpiperazines for Gastrointestinal Cancer Therapy"

_molecules, 2019, doi:10.3390/molecules24203661_

Round 1
Reviewer 1 Report
The manuscript describes synthesis of several indolylpyrimidylpiperazine derivatives as cytotoxic agents. Synthesis is rather trivial and often not very efficient but biological evaluations seem interesting, although I am not an expert in this field. In my opinion this work deserves publication in Molecules, However some modifications are necessary in the synthetic part.
Many sentences should be rewritten using proper English. For example sentence starting in line 105 …..to give 2-chloro……(9) and 2-chloro….(10) in 23% and 11% yield, respectively. expresions “high boiling point” solvent or base are not necessary. line 116, should be “…dimethylamine…” Comound 5 has nitro group in Figure 2 and chloro group in Scheme 2. Copies of 1H and 13C NMR spectra for, at least, final products should be added to Suplementary Materials.
Author Response
Dear reviewer,
thank ou for your valuable feedbacks on our manuscript.
We have taken into consideration your suggestions and we have 1) rewritten using proper English for the Chemistry section of our Results and Discussion, 2) included the 1H NMR and 13C NMR of the four final mIPP compounds in the supplementary information, 3) replaced the 2nd and 3rd synthetic schemes with the removal of repeated information on the synthesis of compound 8, 4) compared to induced-fit docking of compound 1 with the new compounds, compounds 4 – 6, using the hA3AR homology model and 5) included a PROCHECK analysis of our hA3AR homology model to confirm its quality for docking analysis.
We hope that you will find our amended manuscript sufficiently improved for publication.
Reviewer 2 Report
This manuscript describes an updated scaffold for discovery of adenosine receptor agonists as potential GI cancer therapeutic agents. Positional isomers from converting an amide group to a ketone on a lead compound with A2 receptor selectivity produced an A3 selective lead structure. The rationale for the therapeutic utility of A3 selective agents is sound.
The four structurally similar compounds synthesized are partial agonists of relatively low affinity for A3 receptors in the micromolar range. Cytotoxicity studies against two cancer cell lines also show relatively high IC-50 concentrations. Although the effective concentrations are high, the new scaffold is a good starting point for further discovery.
The chemistry is straightforward and appropriately presented.
Parts of the synthetic schemes described in Scheme 1 are repeated in Schemes 2 and 3. For example, the synthesis of Compound 8 is drawn three times.
TGFalpha shedding assays show that three of the four compounds are selective partial agonists of A3.
The somewhat speculative docking studies are generally reasonable. Details are lacking on the quality of the homology model of the A3 receptor that has been derived from details of the A2 crystal structure. Not necessary, but it would have been useful to compare the results of the lead structure Compound 1 with the new compounds using this model.
Author Response

(The authors gave the same response as above.)
